# The Acromial Index, but Not the Critical Shoulder Angle, Affects Functional and Clinical Outcomes in Patients with Rotator Cuff Tears

**DOI:** 10.3390/diagnostics16010142

**Published:** 2026-01-01

**Authors:** Jin Hyuck Lee, Gyu Bin Lee, Sang Woo Pyun, Woo Yong Chung, Ji Won Wang, Dongik Song, Woong Kyo Jeong

**Affiliations:** 1Department of Sports Medical Center, Anam Hospital, Korea University College of Medicine, Seoul 02841, Republic of Korea; gnkfccc@hanmail.net (J.H.L.); humanwellness@naver.com (G.B.L.); pyun333@gmail.com (S.W.P.); jwy8098@naver.com (W.Y.C.); jiwon-eee@naver.com (J.W.W.); 2Eunpyeong Bone Orthopedics Clinic, 56, Eunpyeong-ro, Eunpyeong-gu, Seoul, Republic of Korea; 10327love@gmail.com; 3Department of Orthopaedic Surgery, Anam Hospital, Korea University College of Medicine, Seoul 02841, Republic of Korea

**Keywords:** rotator cuff muscles tears, critical shoulder angle, acromial index, shoulder muscle performance, University of California at Los Angeles score

## Abstract

**Background/Objective:** This study aimed to compare functional and clinical outcomes in terms of shoulder muscle performance and patient-reported outcomes (PROs) between patients with rotator cuff muscle tears (RCTs) with high preoperative critical shoulder angle (CSA) or acromial index (AI) and those with low preoperative CSA or AI and to determine the outcomes associated with CSA and AI. **Methods:** Ninety patients with RCTs were recruited [45 with high preoperative CSA (>35°) vs. 45 with low preoperative CSA (<35°), and 42 with high preoperative AI (>0.75) vs. 48 with low preoperative AI (<0.75)]. Functional outcomes, such as muscle strength and endurance of the internal rotators, external rotators, and forward flexors, were measured for shoulder muscle performance using an isokinetic device. Clinical outcomes were evaluated using PROs, such as the University of California at Los Angeles (UCLA) and Constant scores. **Results:** Patients with RCTs with high preoperative CSA had decreased muscle endurance of the external rotators (*p* = 0.030) in the involved shoulders compared to patients with RCTs with low preoperative CSA. Patients with RCTs with high preoperative AI had decreased muscle endurance of the external rotators (*p* = 0.010) and UCLA scores (*p* = 0.010) in the involved shoulders compared with patients with RCTs with low preoperative AI. Preoperative AI was closely associated with muscle endurance for external rotators (β = −17.204) and the UCLA score (β = −3.269). **Conclusions:** Patients with RCTs with high preoperative CSA or AI may have lower shoulder muscle endurance than those with low preoperative CSA or AI, especially for external rotators. Furthermore, preoperative AI was independently associated with external rotator muscle endurance and the UCLA score. Therefore, assessment of preoperative CSA or AI may be important for pre- or postoperative management in patients with RCTs, as AI is associated with functional and clinical outcomes.

## 1. Introduction

The most common shoulder injuries are rotator cuff muscle tears (RCTs), which are most commonly associated with symptoms such as pain, limited range of motion (ROM), muscle weakness, and joint dysfunction (neuromuscular control) [1]. RCTs occur due to chronic degenerative or traumatic injuries [2] and are associated with age, physical activities, hypertension, and anatomical structures such as the critical shoulder angle (CSA) and acromial index (AI) [3]. CSA is the angle formed by the glenoid inclination and the lateral extension of the acromion [4], whereas AI is the ratio of the relative lateral protrusion of the acromion to the humeral head [5]. Previous studies have reported that the CSA > 35° [4] or AI > 0.75 [6] is associated with an increased prevalence of RCTs and altered shoulder biomechanics. Therefore, high CSA and AI are associated with the presence of degenerative RCTs [7], which may lead to poorer clinical and functional outcomes.

Previous studies have focused on preoperative fatty degeneration [8,9] and tear size [8] in the rotator cuff muscles in terms of postoperative outcomes. However, Scheiderer et al. have reported that high CSA and AI may increase the risk of retear after rotator cuff repair (RCR), potentially leading to poorer shoulder function [10]. Therefore, several studies have assessed CSA and AI before and after RCR [11,12,13], because high CSA [14] and AI [15] may cause increased muscle injury due to excessive additional force of the rotator cuff muscles and increased shear force of the deltoid muscles to maintain the stability of the shoulder joint [16], which may lead to unfavorable outcomes in terms of shoulder muscle strength. Lee et al. [17] reported that a good preoperative muscle strength ratio of the shoulder internal rotators (IRs) and external rotators (ERs) was associated with better postoperative shoulder function after RCR in patients with RCTs, indicating that preoperative functional status can influence postoperative outcomes. In our clinical experience, patients with RCTs with high preoperative CSA (> 35°) and AI (> 0.75) had a negative impact on preoperative functional outcomes. However, to the best of our knowledge, there are limited available data that directly compare the clinical or functional outcomes among patients with RCTs with high or low preoperative CSA or AI. Thus, it is unclear whether patients with RCTs with high preoperative CSA or AI have worse clinical or functional outcomes than those with RCTs with low preoperative CSA or AI. Furthermore, preoperative CSA and AI may affect rotator cuff muscles and deltoid function, which are also expected to be related to pre- and postoperative functional outcomes; however, available data on the correlations of preoperative CSA, AI, and functional outcomes are lacking.

Therefore, the purpose of this study was to compare functional and clinical outcomes in terms of shoulder muscle performance and patient-reported outcomes (PROs) between patients with RCTs who had high versus low preoperative CSA or AI and to determine the outcomes associated with CSA and AI. We hypothesized that patients with RCTs who had a high preoperative CSA or AI would have worse functional outcomes compared with those with a low preoperative CSA or AI.

## 2. Methods

This retrospective comparative study complied with the Declaration of Helsinki and followed the STROBE guidelines for non-pharmacological treatments. This study was approved by the institutional review board (2014AN0090), and informed consent was obtained from all participants prior to arthroscopic RCR.

### 2.1. Study Design and Patient Enrollment

This study retrospectively analyzed prospectively collected data from 426 patients diagnosed with RCTs between 2015 and 2023. The inclusion criteria were as follows: (1) only full-thickness RCTs that preoperatively underwent all functional evaluations; (2) preoperative true anteroposterior (AP) radiographs to assess CSA and AI. The exclusion criteria were as follows: (1) partial-thickness tears and revision surgery; (2) other lesions, such as superior labrum from anterior to posterior, shoulder dislocation, calcific tendinitis, and stiff shoulder; (3) pseudoparalysis; (4) failure to perform functional evaluation due to pain; and (5) previous shoulder surgery within 1 year, such as due to fractures. In the present study, high CSA and AI were defined as CSA > 35° [18] and AI > 0.75 [6,19]. Of the 426 patients, 336 were excluded; thus, the final assessment used data from 90 patients (45 with high preoperative CSA vs. 45 with low preoperative CSA, and 42 with high preoperative AI vs. 48 with low preoperative AI) (Figure 1). The CSA and AI groupings overlap (the same patients may be included in both comparisons). Among the 90 patients, 26 (28.9%), 50 (55.5%), and 14 (15.6%) had small, medium, and large tears, respectively. The tear size of the rotator cuff muscles was classified as small (<1 cm), medium-sized (1–3 cm), or large to massive (>3 cm). Furthermore, 41 (45.6%) patients had subscapularis tears, which were classified using the Yoo and Rhee classification [20]. Fatty infiltration was confirmed using the Goutallier classification [21]. All patients simultaneously underwent imaging and functional assessment during outpatient visits.

### 2.2. Radiographic Assessment

Based on previous studies [6,18,19], preoperative CSA and AI were measured on true anteroposterior (AP) shoulder radiographs. True AP radiographs were evaluated according to the Suter–Henninger classification [22], considering scapular rotation, scapular tilt, and beam direction. Radiographs showing excessive malrotation or scapular tilt were excluded from analysis. Glenoid version or inclination was not assessed. CSA and AI were measured by three independent assessors, including radiologists, with >4 years of experience. The intraclass correlation coefficients (ICCs) were excellent for CSA (0.94) and AI (0.92), and the mean of three measurements was used for analysis.

### 2.3. Functional Outcomes

#### Shoulder Muscle Performance

Shoulder muscle strength and muscle endurance were measured for IRs, ERs, and forward flexors (FFs) using a quantified isokinetic device (Biodex multi-joint system 4, Biodex Medical System, Shirley, NY, USA) [23]. All tests were performed on both shoulders with the patient seated in an upright position. For the rotational strength test, internal and external rotation were performed with the shoulder positioned at 30° of abduction and the elbow flexed at 90°. For the forward flexion strength test, the movement was performed in the scapular plane with the elbow fully extended. Before testing, all patients performed a warm-up protocol consisting of shoulder stretching exercises and 5–8 repetitions at a submaximal load.

Peak torque/body weight (PT/BW, Nm kg^−1^ × 100) and total work (TW, J) were regarded as muscle strength and muscle endurance, respectively [23]. PT/BW was recorded during five repetitions at 60°/sec, and TW was recorded during fifteen repetitions at 180°/sec. In the present study, the ICCs for IRs, ERs, and FFs were 0.90, 0.92, and 0.88, respectively. In addition, a coefficient of variation (CV, %) < 15% is commonly indicated as acceptable reliability in the test [24,25], and the CV for IRs, ERs, and FFs was 9.6, 12.7, and 13.2, respectively, in the present study.

### 2.4. Clinical Outcomes

#### Patient-Reported Outcomes (PROs)

PROs were assessed for shoulder function using the University of California at Los Angeles (UCLA) and Constant scores [11,19]. Higher scores were related to good clinical status. The UCLA and Constant scores were administered in Korean. The questionnaires were translated and explained to the patients by a trained clinician to facilitate comprehension. Clinical assessments were performed by clinicians who were independent of those performing the radiographic measurements.

### 2.5. Statistical Analysis

Based on the findings of a previous study [19], a difference in Constant scores > 16 points between patients with preoperative AI < 0.75 and preoperative AI > 0.75 was regarded as clinically significant. Therefore, an a priori power analysis was used to determine the sample size, with a power and α level of 0.8 and 0.05, respectively. A total of 48 patients (effect size d = 0.827) were required to demonstrate a clinically significant difference in Constant scores of >16 points between the groups, and the power was 0.800.

The normal distribution and assumption of equal variance were verified using the Shapiro–Wilk and Levene tests, respectively. Continuous and categorical variables were compared using independent t-tests and chi-square tests, respectively. Independent t-tests were used to compare the means of the independent variables between the groups (CSA > 35° vs. CSA < 35°, and AI > 0.75 vs. AI < 0.75), with Bonferroni correction applied for multiple comparisons. In addition, effect sizes were calculated as Cohen’s d using the group means and pooled standard deviations. All continuous variables are presented as mean ± standard deviation. Pearson’s correlation analysis was used to assess associations between preoperative CSA or AI and demographic characteristics, shoulder muscle performance, and patient-reported outcomes (PROs) in the entire cohort of patients (n = 90). Variables demonstrating significant correlations were considered candidates for multivariable analysis. Multiple linear regression analyses were performed to identify independent associations between CSA and AI and selected outcome measures. CSA and AI were simultaneously included as independent variables, with age, sex, tear size, and fatty degeneration entered as covariates in all models, regardless of their statistical significance, to control for potential confounding effects. To evaluate the interaction between CSA and AI, an additional analysis was performed within the multivariable regression model. Statistical analyses were performed using SPSS software (version 21.0; SPSS Inc., Chicago, IL, USA), and statistical significance was set at *p* < 0.05.

## 3. Results

### 3.1. Demographic Data

A total of 90 patients participated in the present study (45 with high preoperative CSA vs. 45 with low preoperative CSA and 42 with high preoperative AI vs. 48 with low preoperative AI). The mean CSA in 45 patients with RCTs with a high preoperative CSA (CSA > 35°) was 40.7° ± 4.3, and the mean CSA in 45 patients with RCTs with a low preoperative CSA (CSA < 35°) was 31.6° ± 2.3. The mean AI in 42 patients with RCTs with a high preoperative AI (AI > 0.75) was 0.84 ± 0.07, and the mean AI in 48 patients with RCTs with a low preoperative AI (AI < 0.75) was 0.66 ± 0.06. Table 1 presents the demographic data. There were no significant differences in sex, age, height, weight, body mass index, dominant and involved side arms, pain, subscapularis tears, tear size, or fatty degeneration between the two groups (all *p* > 0.05) (Table 1).

### 3.2. Comparison of Shoulder Muscle Performance and PROs for CSA

There were no significant differences in IRs, ERs, and FFs between the two groups (all *p* > 0.05), indicating that muscle strength was not significantly different between the groups. TW was significantly different in ERs between the two groups (95% confidence interval [CI]: −55.7, −3.0, *p* = 0.030) but not in IRs and FFs (all *p* > 0.05), indicating that patients with RCTs with high preoperative CSA had decreased muscle endurance for ERs in the involved shoulders compared to patients with RCTs with low preoperative CSA. PROs, such as UCLA and Constant scores, were not significantly different between the two groups (all *p* > 0.05) (Table 2).

### 3.3. Comparison of Shoulder Muscle Performance and PROs for AI

There were no significant differences in the IRs, ERs, and FFs between the two groups (all *p* > 0.05), indicating that muscle strength was not significantly different between the groups. TW was significantly different in ERs between the two groups (95% CI: −60.1 to −8.3, *p* = 0.010) but not in IRs and FFs (all *p* > 0.05), indicating that patients with RCTs who had a high preoperative AI exhibited decreased muscle endurance for ERs in the involved shoulders compared to those who had a low preoperative AI. The UCLA score was significantly lower in patients with RCTs who had a high preoperative AI than in those who had a low preoperative AI (95% CI: −4.9 to −0.7, *p* = 0.010), but not the Constant score (*p* > 0.05) (Table 2).

**Table 2 diagnostics-16-00142-t002:** Shoulder muscle performance and patient-reported outcomes between the groups.

	CSA > 35° (n = 45)	CSA < 35° (n = 45)	*p*-value	95% CI
				*Cohen’s d*
IR strength	31.5 ± 11.1	32.9 ± 12.3	0.585	−6.3 to 3.6
				−0.11
ER strength	13.0 ± 5.7	14.8 ± 6.0	0.152	−4.2 to 0.7
				−0.30
FF strength	32.1 ± 12.6	32.2 ± 13.0	0.450	−5.4 to 5.2
				0
IR endurance	279.4 ± 127.3	296.0 ± 180.0	0.089	−121.8 to 8.8
				−0.10
ER endurance	70.8 ± 50.8	100.1 ± 73.0	**0.030 ***	−55.7 to −3.0
				−0.47
FF endurance	262.1 ± 196.9	290.0 ± 161.4	0.450	−104.2 to 46.6
				−0.15
UCLA score	19.6 ± 6.1	20.1 ± 4.1	0.643	−2.7 to 1.7
				−0.10
Constant score	50.9 ± 15.2	50.0 ± 16.2	0.766	−5.6 to 7.6
				0.06
	AI > 0.75 (n = 42)	AI < 0.75 (n = 48)	*p*-value	95% CI
				*Cohen’s d*
IR strength	31.2 ± 10.6	32.4 ± 12.1	0.613	−6.0 to 3.6
				0.11
ER strength	12.8 ± 5.1	15.1 ± 6.3	0.063	−4.7 to 0.1
				−0.40
FF strength	33.6 ± 12.4	34.5 ± 16.9	0.769	−7.2 to 5.3
				−0.06
IR endurance	277.7 ± 131.4	298.5 ± 177.1	0.130	−116.9 to 15.3
				−0.13
ER endurance	68.6 ± 41.9	102.8 ± 74.8	**0.010 ***	−60.1 to −8.3
				−0.56
FF endurance	289.8 ± 201.4	294.8 ± 182.0	0.903	−85.3 to 75.4
				−0.02
UCLA score	18.5 ± 5.7	21.3 ± 4.3	**0.010 ***	−4.9 to −0.7
				−0.55
Constant score	54.0 ± 13.4	53.5 ± 16.9	0.884	−6.0 to 6.9
				0.03

CSA, critical shoulder angle; AI, acromial index; IR, internal rotator; ER, external rotator; FF, forward flexor; UCLA, University of California at Los Angeles score. Note: The values are expressed as mean ± standard deviation. *** Statistically significant**.

### 3.4. Multiple Linear Regression Analysis for CAS and AI

Correlation analyses were performed between demographic data, functional outcomes, and preoperative CSA or preoperative AI. Bivariate analysis revealed that preoperative CSA was negatively correlated with TW for IRs (r = −0.440, *p* = 0.023) and ERs (r = −0.490, *p* = 0.006); however, preoperative AI was negatively correlated with TW for IRs (r = −0.239, *p* = 0.023), ERs (r = −0.329, *p* = 0.002), and UCLA scores (r = −0.243, *p* = 0.021) (Table 3). For these parameters, multiple linear regression analysis demonstrated that TW for ERs (β = −17.204, *p* = 0.032) and UCLA score (β = −3.269, *p* = 0.039) were independently associated with preoperative AI, but not the CSA (*p* > 0.05) (Table 4).

### 3.5. Interaction Between CSA and AI

The interaction between CSA and AI was not statistically significant for any outcome measures (all *p* > 0.05).

## 4. Discussion

The most important finding of this study was that although muscle strength did not differ between patients with RCTs with high or low preoperative CSA or AI, muscle endurance for ERs was worse in patients with RCTs with high preoperative CSA or AI than in those with RCTs with low preoperative CSA or AI. The UCLA score was worse in patients with RCTs with a high preoperative AI than in those with RCTs with a low preoperative AI; however, the CSA did not differ significantly. Furthermore, preoperative AI was closely associated with muscle endurance in the ERs and UCLA scores.

Several studies [26,27] have reported that the morphological structures of the acromion can lead to impingement during shoulder motion, which can cause RCTs and affect pain and shoulder function, such as strength. However, in the present study, muscle strength did not differ between patients with RCTs with high or low preoperative CSA or AI; however, muscle endurance for ERs was worse in patients with RCTs with high preoperative CSA or AI than those with low preoperative CSA or AI. Although the reason for the results of this study is unclear, they may be explained by compensatory activation in other muscles, such as the deltoid, pectoralis, and periscapular muscles. The deltoid and pectoralis muscles contribute to internal rotation, external rotation, and forward flexion in shoulder joint motion and show compensatory activity patterns to improve dynamic joint stability during shoulder joint motion in patients with RCTs [28,29,30]. Oh et al. [29] reported that in patients with RCTs, the deltoid and periscapular muscles exert appropriate coupling forces to compensate for rotator cuff muscles’ dysfunction. In addition, several studies have reported that as the size of the RCTs increases, the compensatory activity of the deltoid muscles increases because the concavity-compression force decreases [28]. Hence, in the present study, there may be no difference in muscle strength between the groups due to the compensatory activity of these shoulder muscles. However, the deltoid and periscapular muscles may experience fatigue during repetitive movements, resulting in decreased muscle endurance. In particular, as tears extend posteriorly or as subscapularis tears occur, the activity of the posterior muscles may increase more than that of the anterior muscles to maintain shoulder joint stability from the anterior translation of the humeral head [31,32], which may further induce weakness in muscle endurance in the posterior muscles due to muscle fatigue. Several studies have reported that the posterior deltoid [28,30] and trapezius muscles [33] are overactivated in symptomatic patients with RCTs compared to asymptomatic patients with RCTs or normal controls. In addition, Bedeir and Grawe [6] reported that patients with RCTs with a high preoperative CSA and AI (CSA >39° and AI >0.75) had more posterosuperior tears than those with anterosuperior tears. Similarly, Smith and Liu [34] reported that a high preoperative CSA (>37°) was associated with posterosuperior RCTs. Hence, posterosuperior RCTs may result in the weakening of ERs, which may further increase the compensatory mechanisms for shoulder stability. In the present study, patients with RCTs with high preoperative CSA and AI were 40.7° and 0.84, respectively, and approximately 85% of all patients had supraspinatus and infraspinatus tears or teres minor tears. Taken together, the results of this study may explain why there was only a difference in ER muscle endurance. However, this compensatory mechanism is not completely understood.

Contrary to our expectations, in patients with RCTs, muscle strength seems to be influenced by other factors, such as tear size and fatty degeneration, rather than the morphological structures of the acromion. For this reason, in the present study, although there was no difference in tear size and fatty degeneration between the groups, both preoperative CSA and AI were negatively correlated with ER muscle endurance and were not statistically correlated with muscle strength. In particular, preoperative AI was a significant and independent predictor of muscle endurance in ERs. However, the basis for these results remains unclear. Therefore, further high-quality randomized studies are necessary to clarify the results of this study.

Several studies have analyzed UCLA and Constant scores between patients with RCTs with preoperative CSA > 35° and CSA < 35° [11,19], and researchers have reported that both the UCLA and Constant scores were not different between the two groups, which is consistent with our findings. Furthermore, Lee et al. [11] reported that the UCLA and Constant scores were not different between patients with RCTs with preoperative AI > 0.7 and AI < 0.7. However, Opsomer et al. [19] reported that patients with RCTs with preoperative AI > 0.75, but not the UCLA score, had lower Constant scores than patients with RCTs with preoperative AI < 0.75. Our findings differ from those of previous studies. In the present study, patients with RCTs with preoperative AI > 0.75 had higher UCLA scores than those with RCTs with preoperative AI < 0.75, but not Constant scores. Although the reason for this difference from the results of previous studies is unclear, one possible reason may be the difference in the participants. In the present study, the participants had multiple tears in the rotator cuff muscles, such as the supraspinatus, infraspinatus, teres minor, and subscapularis, whereas the participants had only a single tear in the supraspinatus muscle in previous studies. AI is known to be associated with a higher incidence of degenerative RCTs [19], and preoperative tear size is associated with shoulder function and satisfaction [35]. In particular, the UCLA score consists of pain, function, ROM, ER strength, and satisfaction, which is an important tool for evaluating the function of patients with RCTs [18,35]. Hence, the UCLA score may be worse in patients with RCTs with preoperative AI > 0.75 than in patients with RCTs with preoperative AI < 0.75. In the present study, preoperative AI was a significant and independent predictor of the UCLA score. Therefore, to assess the subjective and objective status of patients preoperatively, we should evaluate preoperative AI rather than preoperative CSA.

### Limitations

This study had several limitations. First, only preoperative data, such as CSA and AI, and functional and clinical outcomes were analyzed. Therefore, further studies are needed to compare and analyze the preoperative and postoperative CSA, AI, and functional and clinical outcomes. Second, most participants were female (approximately 70%); thus, caution is needed in interpreting the results owing to gender bias. Third, the glenoid shape was not evaluated [3], which may have affected the CSA. Fourth, the results of this study cannot be generalized because the status of RCTs and muscle strength may be related to age. Finally, patients in the CSA and AI groups overlap; hence, further studies may be needed to clarify the results of this study. Nevertheless, to the best of our knowledge, this study is the first to compare functional and clinical outcomes between patients with RCTs with high and low preoperative CSA and AI. Based on the results of this study, patients with RCTs who have a preoperative CSA > 35° or AI > 0.75 should focus more on ER endurance training to achieve favorable postoperative functional and clinical outcomes, particularly those with an AI > 0.75.

## 5. Conclusions

Patients with RCTs with high preoperative CSA or AI may have lower shoulder muscle endurance than those with low preoperative CSA or AI, especially for the ERs. Furthermore, preoperative AI was a predictor of ER muscle endurance and the UCLA score. Therefore, it is important to assess preoperative CSA or AI for pre- or postoperative management in patients with RCTs, as AI may affect functional and clinical outcomes.

## Figures and Tables

**Figure 1 diagnostics-16-00142-f001:**
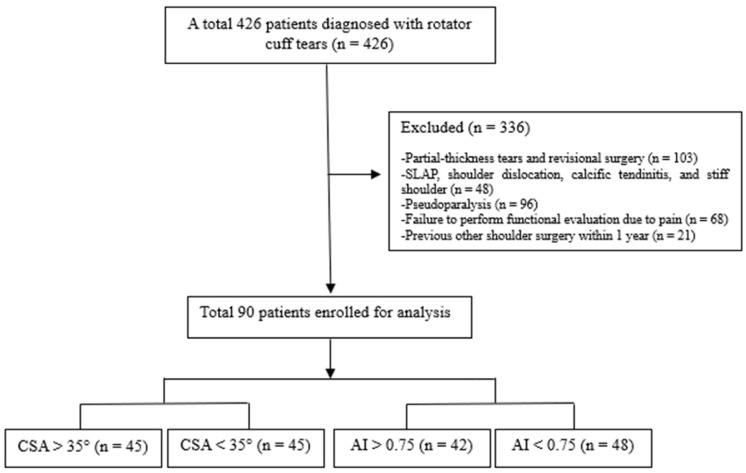
Study flow diagram.

**Table 1 diagnostics-16-00142-t001:** Characteristics of participants.

	CSA > 35° (n = 45)	CSA < 35° (n = 45)	*p*-value
Sex (male/female)	11/34	16/29	0.358
Age (y)	65.5 ± 8.5	67.0 ± 8.0	0.644
Height	158.9 ± 8.6	160.8 ± 8.7	0.281
Weight	63.0 ± 9.2	64.0 ± 11.9	0.647
Body mass index (kg/m^2^)	24.9 ± 3.2	24.6 ± 2.6	0.525
Injured arm (right/left)	38/7	30/15	0.085
Dominant arm (right/left)	45/0	44/1	1.0
Visual Analogue Scale (pain)	6.8 ± 0.3	6.7 ± 0.1	1.0
Subscapularis tears (n: I, IIA, IIB, III, IV, and V)	8, 7, 5, 0, 1,0	9, 8, 3, 0, 0	0.943
Tear size	2.3 ± 1.6	2.1 ± 0.9	0.919
Fatty degeneration (n: O, I, II, III, and IV)			
Supraspinatus	13, 11, 12, 9, 0	14, 5, 20, 4, 2	0.084
Infraspinatus	15, 20, 5, 5, 0	27, 12, 4, 2, 0	0.078
Teres minor	40, 5, 0, 0, 0	36, 7, 2, 0, 0	0.280
Subscapularis	33, 7, 5, 0, 0	40, 4, 1, 0, 0	0.125
	AI > 0.75 (n = 42)	AI < 0.75 (n = 48)	*p*-value
Sex (male/female)	11/31	16/32	0.262
Age (y)	66.7 ± 8.3	65.4 ± 8.5	0.475
Height	159.2 ± 8.4	162.5 ± 8.4	0.065
Weight	61.7 ± 8.2	64.7 ± 9.3	0.105
Body mass index (kg/m^2^)	24.7 ± 3.2	25.0 ± 2.7	0.643
Injured arm (right/left)	32/10	36/12	0.811
Dominant arm (right/left)	42/0	47/1	1.0
Visual Analogue Scale (pain)	6.5 ± 0.2	6.9 ± 0.1	0.297
Subscapularis tears (n: I, IIA, IIB, III, IV, and V)	10, 8, 5, 0, 0	7, 7, 3, 1, 0	0.680
Tear size	2.2 ± 1.9	2.0 ± 0.8	0.767
Fatty degeneration (n: O, I, II, III, and IV)			
Supraspinatus	13, 8, 14, 7, 0	14, 8, 18, 6, 2	0.751
Infraspinatus	15, 19, 4, 4, 0	27, 13, 5, 3, 0	0.141
Teres minor	38, 3, 1, 0, 0	38, 9, 1, 0, 0	0.271
Subscapularis	31, 6, 5, 0, 0	42, 5, 1, 0, 0	0.133

Note: The values are expressed as mean ± standard deviation.

**Table 3 diagnostics-16-00142-t003:** Correlations between parameters, CSA, and AI (n = 90).

	CSA		AI	
Parameters	PCC (r)	*p*-Value	PCC (r)	*p*-Value
IR endurance	−0.440	**0.023 ** *****	−0.239	**0.023 ** *****
ER endurance	−0.490	**0.006 ** *****	−0.329	**0.002 ** *****
UCLA score	-	-	−0.243	**0.021 ** *****

CSA, critical shoulder angle; AI, acromial index; PCC, Pearson correlation coefficient; IR, internal rotator; ER, external rotator; UCLA, University of California at Los Angeles score. *** Statistically significant**.

**Table 4 diagnostics-16-00142-t004:** Multiple linear regression analysis for the associations of CSA and AI with ER endurance and UCLA score.

Dependent Variables	Predictor	Standardized Coefficients				
		B	Error	β	*t*	*p*-Value
ER endurance	CSA	−3.211	1.624	−0.287	−1.978	0.053
	AI	−17.204	7.920	−0.531	−2.172	0.032
	Sex	−47.566	13.904	−0.341	−3.421	0.001
	Age	−1.498	0.770	−0.193	−1.946	0.055
	Tear size	−1.035	5.101	−0.020	−0.203	0.840
	Goutallier SSP	9.503	10.154	0.166	0.936	0.352
	Goutallier ISP	−7.427	17.975	−0.106	−0.413	0.681
	Goutallier TM	2.167	23.977	0.014	0.090	0.928
	Goutallier SSC	2.834	20.175	0.024	0.140	0.889
UCLA score	CSA	−1.466	1.470	−0.161	−0.997	0.322
	AI	−3.269	1.591	−0.556	−2.055	0.039
	Sex	−1.395	1.252	−0.124	−1.114	0.269
	Age	−0.005	0.069	−0.008	−0.077	0.939
	Tear size	−0.140	0.459	−0.034	−0.305	0.762
	Goutallier SSP	−0.643	0.915	−0.139	−0.703	0.484
	Goutallier ISP	2.092	1.619	0.370	1.292	0.200
	Goutallier TM	−4.798	2.160	−0.373	−2.222	0.029
	Goutallier SSC	0.828	1.817	0.086	0.456	0.650

CSA, critical shoulder angle; AI, acromial index; ER, external rotator; UCLA, University of California at Los Angeles; SSP, supraspinatus; ISP, infraspinatus; TM, teres minor; SSC, subscapularis. Note: Model statistics (ERs endurance, R^2^ = 0.270, Adjusted R^2^ = 0.188, F = 3.288, *p* = 0.002, Durbin–Watson = 1.93). Model statistics (UCLA score, R^2^ = 0.095, Adjusted R^2^ = 0.007, F = 0.928, *p* = 0.505, Durbin–Watson = 1.72).

## Data Availability

Data presented in this study are available upon request from the corresponding author.

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
