# Peer review of "The Acromial Index, but Not the Critical Shoulder Angle, Affects Functional and Clinical Outcomes in Patients with Rotator Cuff Tears"

_diagnostics, 2026, doi:10.3390/diagnostics16010142_

Round 1

Reviewer 1 Report

Comments and Suggestions for Authors

Dear Authors,

I congratulate you for this well described study with new findings reported regarding shoulder functionality pre-operative.

I recommend you to describe the Critical Shoulder Angle and the Acromial Index in the Introduction instead in Method chapter and to add some interpretations of the values from specialized literature in order to be easier to the reader to follow the goals of the study. Also, I recommend at the Limitation chapter to define the expression,  ,,should pay attention to ERs” or to replace that with the meaning, for example, to train more the ERs endurance “ 

Best regards

Author Response

Thank you for the review and comments. Your comments have greatly improved the quality of our study. We have made further editorial improvements to ensure our paper is as publication ready as possible.

Please see the attatched file. 

Reviewer 2 Report

Comments and Suggestions for Authors

Dear Authors,

The study effectively compared functional and clinical outcomes in patients with different shoulder angles by utilizing both objective (isokinetic test) and subjective (PROs) indices simultaneously. Important clinical variables, such as tear size, were well controlled between groups, enhancing comparability.

The Methods section indicates overlap between the CSA and AI groups; however, this overlap was not accounted for in the statistical analyses. It is advisable to use multivariate statistical methods, such as logistic regression or mixed models.

Most participants were female (approximately 70%). A subgroup analysis by gender should be conducted, or the gender bias should be clearly addressed in the interpretation of the results.

Although tear size (small, medium, large) was recorded, it was not included as a covariate in the final analyses (multiple regression).

The difference in UCLA scores across AI groups is significant, but is this difference (approximately 2.8 points) clinically meaningful?

If there is a long-time interval between imaging and functional assessment, the patient's condition may have changed. The authors didn’t state the time.

There is a failure to report the interaction effect between CSA and AI variables. Since the CSA and AI indices are anatomically related, their interaction effect may be significant.

Best regards.

Author Response

(The authors gave the same response as above.)

Reviewer 3 Report

Comments and Suggestions for Authors

1) More emphasis could be placed on the literature showing the relationship between preoperative muscle function and postoperative outcomes (e.g., preoperative strength ratios, ER/IR ratio), and what the current study adds to this could be more clearly highlighted. 2) The abbreviation “RCms” and the concepts of “rotator cuff muscles” / “rotator cuff tears (RCTs)” should be used more consistently (they are sometimes confused). 3) CONSORT stands for randomized controlled trials; a reference to STROBE (observational studies) would be more appropriate here. This sentence should be rewritten. 4) The overlap of the groups should be clearly stated: The same patient is included in both the CSA high/low analysis and the AI ​​high/low analysis. This should not be perceived as two independent cohorts. The methods section should clearly state: “The CSA and AI groupings overlap (the same patients may be included in both comparisons).” 5) The sentence “and it the average of 3 assessors” is grammatically incorrect; The sentence should be corrected to: “The mean of three measurements was used for analysis.” The criteria for True AP radiography (scapula rotation, tilt, beam direction) could be described more clearly; this is important for measurement reliability. Glenoid shape was not evaluated; this is given as a limitation in the discussion, but it could be briefly noted in the methods section (e.g., “Glenoid version or inclination was not assessed.”). 6) Missing/Confusing Points: Which shoulder(s) were measured? Only the “involved shoulder,” or was the opposite side also measured as a reference? The text mostly mentions the “involved shoulder,” but the methods section should clarify this. Measurement Position: Sitting/semi-reclining position, shoulder abduction angle, elbow flexion, forearm position, etc., should be described in more detail. The warm-up protocol (number of repetitions of submaximal trials) is not specified. Those who could not tolerate the test due to pain were already excluded, but was the pain score recorded during the test? If acquired, it may concomitantly affect muscle performance. 7) The language version and validation used should be specified (Korean, English, what is the patients' native language?). Who performed the assessment? The blinding status (independent of those performing radiographic measurements?) should be explained. 8) Multiple comparisons: Numerous t-tests are performed for many muscle parameters (IR, ER, FF; strength and endurance) + two groups (CSA, AI), but multiple test correction (Bonferroni, Holm, etc.) has not been applied. 9) Regression analysis: Text: “multiple linear regression analysis was performed to identify variables that independently affected the CSA or AI in the involved shoulders.” However, the result written in 3.4 is, “AI is a predictor of ER endurance and UCLA score.” Table 4 also shows CSA/AI as the dependent variable and IR/ER endurance and UCLA as the independent variables. At this point, there is a mismatch between the text and the table: Either the table needs to be rearranged (dependent variables: ER endurance and UCLA; independent variables: CSA and AI), Or the text and results need to be rewritten. 10) Some p-values ​​and averages do not seem realistic: e.g., p=1.0 is given when VAS averages are very close; this needs to be clarified whether it is rounding or a calculation error. There is repetition in the Fatty degeneration row, such as “â…¡, â…¡, â…¢, â…£”, probably a typographical error. 11) Cohen’s d = 9.67 is reported for ER endurance in the CSA groups; this is an impossibly large effect biologically (in practice, even d>3 is very rare). This is probably a typesetting error (it should be “-0.67” or similar). This must be corrected. The signs and magnitudes for Cohen’s d should be recalculated in all rows; The Methods section should specify how d was calculated (Hedges g or pooled SD). The numerical magnitude can be emphasized in the conclusions. 12) In Table 3, some p-values ​​appear inconsistent with the correlation magnitudes (e.g., p=0.023 for r=-0.539). This p-value should be much smaller considering the number of n; the correlation analysis should be rechecked. The table title is unclear about which group (CSA high/low or the entire cohort) it belongs to; it is likely that the correlation was performed for all 90 patients, this should be clearly stated in the text. The regression table should be redesigned as mentioned above; Suggestion: Model 1: ER endurance (dependent) ~ AI + CSA + age + gender + tear size + fatty degeneration Model 2: UCLA score (dependent) ~ AI + CSA + same covariates The regression report should also include information such as standard error, p, R², and model F-statistic along with β. 13) There are numerous minor grammatical and typographical errors: “retrospectively reviewed prospectively collected the data” “and it the average of 3 assessors” “RCTs patients” instead of “patients with RCTs” Singular/plural agreement in the sentence “RCTs patients with a high preoperative AI had decreased…”

Comments on the Quality of English Language

There are numerous minor grammatical and typographical errors:

Author Response

(The authors gave the same response as above.)

Round 2

Reviewer 2 Report

Comments and Suggestions for Authors

Dear Authors,

You successfully addressed the comments from the primary review. I think the manuscript quality is now suitable for publication.

Best regards.

Reviewer 3 Report

Comments and Suggestions for Authors

I think it can be published in its current form. The necessary corrections appear to have been made.